# Perceived Social Support and Quality of Life of Children with and without Developmental Disabilities and Their Caregivers during the COVID-19 Pandemic in Brazil: A Cross-Sectional Study

**DOI:** 10.3390/ijerph20054449

**Published:** 2023-03-02

**Authors:** Isabelle Gansella Rocha Da Costa, Beatriz Helena Brugnaro, Camila Resende Gâmbaro Lima, Olaf Kraus de Camargo, Lais Fumincelli, Silvia Letícia Pavão, Nelci Adriana Cicuto Ferreira Rocha

**Affiliations:** 1Child Development Analysis Laboratory (LADI), Department of Physical Therapy, Federal University of São Carlos (UFSCar), São Carlos 13565-905, SP, Brazil; 2CanChild, Department of Pediatrics, McMaster University, Hamilton, ON L8S 1C7, Canada; 3Department of Nursing, Federal University of São Carlos (UFSCar), São Carlos 13565-905, SP, Brazil; 4Department of Prevention and Rehabilitation in Physical Therapy, Federal University of Paraná (UFPR), Curitiba 80060-000, PR, Brazil

**Keywords:** COVID-19, children, disabilities, family, quality of life, social support

## Abstract

Background: Social support and Quality of life (QoL) are important aspects of life and should be explored during the specific scenario of the pandemic. Aims: (i) to compare the perceived social support (PSS) in caregivers and the domains of QoL of the caregiver and the child with developmental disabilities (DD) and typical development (TD); (ii) to verify the existence of the association, in each group, between the PSS, and the domain of QoL of the caregiver and the child. Methods and Procedures: 52 caregivers of children with DD and 34 with TD participated remotely. We assessed PSS (Social Support Scale), children’s QoL (PedsQL-4.0-parent proxy) and caregivers’ QoL (PedsQL-Family Impact Module). The groups were compared for the outcomes using the Mann–Whitney test, and Spearman’s test evaluated the correlation between the PSS and the QoL (child and caregiver) in each of the groups. Outcomes and Results: There was no difference between groups for PSS. Children with DD presented lower values in PedsQL total, psychosocial health, physical health, social activities, and school activity. Caregivers of children with TD presented lower values in PedsQL family total, physical capacity, emotional aspect, social aspect, daily activities, and higher value in communication. In the DD group, we found a positive relationship between PSS with child: Psychosocial Health (r = 0.350) and Emotional Aspect (r = 0.380), and with family: Total (r = 0.562), Physical Capacity (r = 0.402), Emotional Aspect (r = 0.492), Social Aspect (r = 0.606), Communication (r = 0.535), Concern (r = 0.303), Daily Activities (r = 0.394) and Family Relationships (r = 0.369). In the TD group, we found that PSS was positively associated with Family: Social Aspect (r = 0.472) and Communication (r = 0.431). Conclusions and Implications: During the COVID-19 pandemic, despite both groups presenting similar PSS, there are important differences in QoL between them. For both groups, greater levels of perceived social support are associated with greater caregiver-reported in some domains of the child’s and caregiver’s QoL. These associations are more numerous, especially for the families of children with DD. This study provides a unique view into the relationships between perceived social support and QoL during the “natural experiment” of living through a pandemic.

## 1. Introduction

Primary home care of a child with developmental disabilities (DD) usually requires changes in the physical, psychological and social spheres of the family, resulting in higher physical, emotional, and even financial burdens compared to parents of children with typical development (TD). It can lead to chronic stress and trigger physical and psychological disorders for the caregiver, which may reflect on their and the child’s quality of life (QoL) [1,2].

Although caring for a child with DD can have positive effects on parents’ lives, such as personal growth, family unity and proximity, and greater sensitivity and tolerance towards diversity [3], several factors can be stressors inherent in the care and the routine. We highlight the loss of support of people and social stigma [4], changes in the routine of daily life with social restrictions [5], stress [6] and concerns about the health and development of the child [7]. Because of these greater demands, these families often need social support [8].

Social support is defined as the perceived availability or momentary level of receipt of social contacts to fulfill specific functions [9], and it is an important indicator of social relationships [10]. There are two forms of social support: informal, provided by family, friends, and other close people, and formal, provided by professionals, social, and health institutions [11]. In addition, formal or informal social support can promote and be related to health since it incentivizes health behaviors [12].

An important context that changed social relationships and people’s lives around the world was the COVID-19 pandemic [13,14]. In Brazil, while creating strategies against the pandemic, services such as schools, healthcare centers, and leisure and recreation centers were closed, in addition to prohibiting access to places and events that promoted overcrowding [15]. In this way, this new scenario of life drastically changed a family’s routine, from declines in their child’s motor, behavior, social, and communication skills to changes in therapy services [16]. Therefore, the perceived social support and the QoL gain even more attention in this context and should be explored.

Studies published before the pandemic have shown that health-related social support and QoL are positively associated in groups of parents of children with congenital malformations [17] and that the perceived social support of caregivers is related to fewer depressive symptoms [18]. One study reported that a high level of social support is related to a lower incidence of depression and parental stress in parents of children with cerebral palsy [19]. Also, receiving social support has a strong association with the family’s quality of life [20]. The importance of this relationship for the families of children with DD is remarkable.

Considering the specific context of the COVID-19 pandemic, there are studies indicating that caregivers of children with special healthcare needs received inadequate social support during the second wave of COVID-19 [21]. A qualitative study showed that informal support was an important factor for caregiving demands and resources for coping, according to parents of children with intellectual disabilities [22]. Also, the pandemic limited access to social support and services, impacting the mental health of children with disabilities and their parents [23,24,25].

Studies addressing the pandemic period have shown that caregiver concerns for the child are associated with caregivers’ symptoms of stress, depression, anxiety and behavioral and emotional deterioration during the lockdown [26,27,28,29]. In addition, studies have indicated an increase in demands on caregivers of children and youth with autism during the pandemic globally [23] and a greater burden of care for mothers who have a handicapped child [30]. A qualitative study of people with autism and caregivers of people with autism identified that the pandemic was related to mental health problems and a lack of access to support [31]. Although studies indicate negative aspects of the pandemic on the health of caregivers of children with disabilities, we did not find studies that evaluated the quality of life, addressing the physical functioning, emotional functioning, social functioning, and functioning in school domains, comparing children with DD and TD. This represents a gap in the literature during the COVID-19 pandemic.

Therefore, our study proposes to evaluate these aspects during the pandemic for several domains related to QoL, aiming to identify in which QoL domains this correlation exists and to allow the identification of more specific areas for evaluations and interventions focused on the family since the pandemic is a potential aggravator of these aspects.

The objectives of this study were, during the COVID-19 pandemic, (i) to compare the perceived social support in caregivers and the QoL in the domains of psychosocial, physical, emotional, social and schools activities of the caregiver and the child with developmental disabilities (DD) and typical development (TD), and (ii) to verify the existence of associations, in each group, between the perceived social support, and the QoL of the caregiver and the child, in the domains of psychosocial, physical, emotional, social and schools activities.

Taking into account other studies previous to the pandemic related to perceived social support and quality of life comparing groups with DD and TD, as well as the changes that the pandemic has brought about in an abrupt and unprecedented way, we expect to find that the perceived social support will be similar in both groups since the challenges of pandemic occurred in all people, and the difficulties in obtaining social support were not a function of having an impairment, but due to the mandated physical distancing. For the QoL, we expect that the DD group (children and caregivers) will show lower QoL when compared to the TD group since the pandemic might exacerbate the challenges for this group. Also, it is expected that greater social support will be associated with a higher quality of life for the caregiver and the child, especially in the group with DD, since caregivers and children with DD often need more social support [8] (Araújo et al., 2016) and the social support and QoL showed to be associated before pandemic [17] (Silva et al., 2020). These results will contribute to a better understanding of these aspects during the pandemic.

## 2. Methods

### 2.1. Study Design

This is an observational study with a cross-sectional design and a convenience sample, which followed STROBE guidelines [32], as well as the recommendations of the Check-list for Reporting Results of Internet E-Surveys (CHERRIES) statement guideline. It was approved by the Ethics Committee on Research in Human Beings of the Federal University of São Carlos (protocol number: 42344221.0.0000.5504).

### 2.2. Sample Calculation

The sample size was determined a priori using the software GPower (Version 3.1.3, University Düsseldorf, Düsseldorf, Germany). Based on the study by Varni et al. (2004) [33], considering the PedsQL Total as an outcome variable, the means of the healthy group and those with some health condition being 87.61 and 74.22, respectively. Thus, an effect size of 0.86 was obtained, and a test power of 90% and significance of 0.05 was used, which resulted in 30 children in each group.

### 2.3. Participants

Fifty-two caregivers of children aged 5–12 years with DD (*n* = 52) and TD (*n* = 34) took part in the study. We performed remote data collection, so we could evaluate participants from all geographic regions of Brazil. Recruitment occurred by dissemination of the research through social media.

To respond to the survey, participants should be the main caregiver of a child with or without DD and need to agree and virtually sign the Informed Consent Form (ICF). The reported diagnoses of children with DD included cerebral palsy, Down syndrome, myelomeningocele, congenital malformations, and Autism Spectrum Disorder, among others, as shown by the flowchart in Figure 1.

### 2.4. Procedures

The participants were recruited through dissemination on social networks, Facebook and Instagram, radio, and email networks, carried out by the communication department of the university in which the study was developed. Data was collected through online Google^®^ forms, between March 2020 and April 2021, at the height of the COVID-19 global pandemic. The results were sent directly to a password-protected Google Drive^®^ of the researchers.

### 2.5. Measures

#### 2.5.1. Social Support Scale

The Social Support Scale (SSS) was used to evaluate the perceived social support during the pandemic period of the caregivers from both groups, TD and DD. SSS is translated and validated for Brazilians, presenting high internal consistency in all its domains [34]. SSS encompasses 19 items that assess 3 domains of social support: emotional, material, and affective/interaction. For each of these items, on a 5-point scale from “never” (score 1); “rarely” (score 2); “sometimes” (score 3); “almost always” (score 4); to “always” (score 5), participants respond to how much they consider that they have that particular social support described [34]. Adding the scores for each item, the higher the gross score, the greater the perceived social support received by the respondent.

#### 2.5.2. Pediatric Quality of Life Inventory (PedsQL) Version 4.0, Parent Proxy

PedsQL is a translated, adapted and valid instrument to measure Health-Related Quality of Life (HRQoL) in Brazilians [35], i.e., QoL related to the health of the individual [33]. Since this instrument is a parent proxy, it can be used in caregivers of children and adolescents aged 2 to 18 years, healthy or with any type of health condition. PedsQL is composed of 23 items divided into 4 different dimensions: physical functioning (8 items); emotional functioning (5 items); social functioning (5 items); and functioning in school (5 items). Items are scored as 0) it never constitutes a problem; (1) it is almost never a problem; (2) sometimes it is a problem; (3) it is often a problem; (4) it is almost always a problem. Although scores vary from 0–4, the final score is converted afterward into a scale of 0 to 100 and follows a decreasing pattern, i.e., the option “never” scores 100 and “almost always” scores 0.

The results can be reported in 3 ways: (a) the sum of both categories (total score of 23 items); (b) results addressing physical health (8 items); (c) results addressing psychosocial health (15 items). To achieve the total final score and domains’ scores separately, each item’s score is converted into a scale from 0 to 100, and then the scores of all items were summed and divided by the number of items [35]. To create the psychosocial health summary score, the mean is computed as the sum of the items over the number of items answered in the emotional, social, and school functioning scales. The physical health summary score is the same as the physical functioning Scale Score. Its completion takes approximately 5 min [36]. For statistical analysis, the total converted scores, physical health, and psychosocial health scores were used, and the higher the score, the better the reported QoL.

#### 2.5.3. PedsQL Family Impact Module (PedsQL-FIM)

PedsQL-FIM evaluates the impact of chronic pediatric health conditions on parents and families. It was translated, adapted and validated for Brazilians [37]. The instrument has 36 items divided into 6 subscales that measure the functioning reported by parents in relation to their: Physical Functioning (6 items); Emotional functioning (5 items); Social Functioning (4 items); Cognitive functioning (5 items); Communication (3 items) and Concern (5 items). In addition to these 6 subscales, there are 2 more that assess caregiver functioning by the parents’ report: Daily Activities (3 items) and Family Relationship (5 items). Its completion takes, on average, 5 min. As well as in PedsQL, each item is scored from 0 to 4, and final scores are converted to a scale from 0 to 100, as occurs in PedsQL v.4.0 described in the item above [33].

The total final score is obtained by adding the converted points of each of the 36 items and dividing the sum by the number of items answered. This score was used in statistical analysis. In the case of subscores, the scores are obtained from the exclusive sum of the specific questions of each domain. The time to complete is approximately 5 min. In all cases, the final score indicates that the more points, the lower the negative impact on the life of this family member [37].

### 2.6. Statistical Analysis

Shapiro-Wilk test revealed the non-normality of the data. Therefore, the Mann–Whitney test compared measures between DD and TD groups. The compared measures were: (a) perceived social support; (b) Family QoL (Total PedsQL Family; PedsQL Physical Ability; PedsQL Emotional Aspect; PedsQL Social Aspect; PedsQL Mental Capacity; PedsQL Communication; PedsQL Concern; PedsQL Everyday Activities; PedsQL Family Relationships); and (c) child’s QoL by the parents’ report (Total PedsQL; PedsQL Psychosocial Health; PedsQL Physical Health; PedsQL Emotional Aspect; PedsQL Social Activities; PedsQL School activity).

Spearman’s correlation tested the association between perceived social support and QoL scores (child and caregiver) in each of the groups. Correlations were classified according to the Cohen and Holliday classification (1982) [38]: up to 0.19: very weak; between 0.20 and 0.39: weak; from 0.40 to 0.69: moderate; from 0.70 to 0.89: strong; and from 0.90 to 1: very strong). For all analyses, a significance level of 5% was adopted.

## 3. Results

### 3.1. Participants

Table 1 illustrates the characteristics of the sample considering groups with TD and DD, regarding sex (female vs. male) and age (years) for children and their caregivers, caregiver situation about work (work in person, work in a home office, do not work), maternal schooling (which increasingly categorizes: Incomplete Elementary, Complete Elementary, Incomplete high school, Complete high school, Incomplete Higher Education, Complete Higher Education), and socioeconomic classification (according to the Brazilian Economic Classification—ABEP (D–E, C1, C2, B1, B2, and A). ABEP addresses socioeconomic aspects based on the number of basic items in one’s house, educational level, availability of running water and paved street to the residence.

### 3.2. Comparison between Groups (DD × TD)

We did not find significant differences in perceived social support between the groups during the pandemic period. Regarding QoL, the children in the DD group presented significantly lower scores than the children with TD in the QoL variables of the Total PedsQL child; PedsQL Psychosocial Health; PedsQL Physical Health; PedsQL Social Activities; PedsQL School Activity; and in the QoL of the PedsQL Family Communication. Differently, children with DD presented higher scores in the variables Total PedsQL Family; PedsQL Family Physical Capacity; PedsQL Family Emotional Aspect; PedsQL Family Social Aspect; and PedsQL Family Daily Activities when compared to the TD. The results of the comparison are found in Table 2 and Table 3.

### 3.3. Correlation Analysis

For the DD group, we found significant correlations between perceived social support and the variables: PedsQL Child Psychosocial Health; PedsQL Child Emotional Aspect; PedsQL Family; PedsQL Family Physical Ability; PedsQL Family Emotional Aspect; PedsQL Family Social Aspect; PedsQL Family Communication; PedsQL Family Concern; PedsQL Family Everyday Activities and PedsQL Family Family Relationships.

For the TD group, significant correlations were found between perceived social support and the variables: PedsQL Family Social Aspect and PedsQL Family Communication. Table 4 illustrates the correlation results within the TD group and the DD group and shows the classification of significant correlations according to Cohen and Holliday (1982) [38].

## 4. Discussion

Our study aimed to compare the perceived social support in caregivers and the QoL in the domains of psychosocial, physical, emotional, social, and school activities of the caregiver and the child with DD and TD in a peculiar scenario of the COVID-19 pandemic. Moreover, for each of these groups, we tested the association between the perceived social support and QoL of children and their caregivers.

### 4.1. Perceived Social Support and Quality of Life

According to our initial expectations, we did not find significant differences in perceived social support between the groups. The reduced levels of social support experienced by families with children presenting DD are well documented in the literature [39,40]. However, during the pandemic, the discrepancies in relation to the availability of social support usually experienced by families of children with DD compared to families of children with TD did not happen. Possibly, due to the scenario of social isolation and restriction of interactions, families of typical children had a reduction in the social support they normally received from others. In turn, parents of children with DD already experience situations of limited social support, mainly restrictions on support from the government and schools [41]. We believe that these limitations remained, which explains the similarity in the results of the two groups.

The assessment of QoL regarding the children resulted in lower values of total QoL, psychosocial health, physical health, social activities, school activities and family communication for children with DD. Previous studies have reported lower QoL scores for children with DD [42,43]. The presence of some physical, motor or cognitive disorder in the child’s development may restrict access to opportunities for activities, recreation, and participation in various life contexts, consequently impacting their QoL [44,45,46]. During the pandemic, these characteristics remained, and the children kept showing reduced levels of physical and psychological health than their typical peers. In addition, the socioeconomic strata of our sample might have contributed to our results. There is a prevalence of lower socioeconomic levels for participants in the DD group. More than 50 percent of the caregivers are part of the socio-economic strata C1–C2 and D–E, which represent, respectively, middle and lower class). Otherwise, caregivers in the TD group are concentrated in classes A, B1 and B2, which represent higher socioeconomic strata. In fact, for children, a caregiver’s socioeconomic level might favor QoL since parents with greater purchasing power would be expected to be able to provide their kids with diverse and enriching experiences, especially related to fun. According to previous studies, a higher socioeconomic stratum can provide better treatment options, complementary therapies, and home adaptations and improve daily life [47,48], which might positively affect the child’s physical and psychosocial QoL, by granting greater comfort related to health and social interactions [49].

Still, regarding QoL, some results were not in accordance with our initial hypotheses. In this unprecedented pandemic scenario, we surprisingly found better caregiver QoL for physical capacity, emotional aspects, social aspects, daily activities, and in Total PedsQL Family for caregivers of children with DD compared to TD. These results may be associated with the peculiarities of the COVID-19 scenario. This is a very interesting result since, based on previously published studies [50,51,52], we initially expected to find a higher QoL for caregivers of the TD group, assuming that these caregivers do not have to deal with the challenges of raising a child with impairments. Nevertheless, our results pointed in the opposite direction. All around the world, families had to deal with the new and unexpected. In a qualitative study addressing mothers of children with multiple health conditions, Pozniak and Kraus de Camargo (2021) [53] reported that COVID-19 had forced society to adopt measures that benefit these caregivers, such as virtual services, supports, and community-building opportunities. These factors might have contributed to determining higher values of QoL for caregivers of children with DD since, for the very first time, these caregivers could participate in activities that they were not able to do before.

Other aspects might have also contributed to our results. Caregivers of children with TD possibly did not present such a high demand for full-time home care. Moreover, concerning the work situation of the caregivers, it was noticed that in the DD group, more than half of the sample was not working in that period, thus being able to devote themselves to the care of the children. While in the TD group, 91.1% of caregivers were working during the pandemic and, thus, had to conciliate their employment with the care of children at home and in remote education. Accordingly, the pandemic context, with the adoption of the remote teaching model for the child [54], may have contributed to caregivers of children with TD exhibiting lower QoL levels compared to the group of DD caregivers.

### 4.2. Relationship between Perceived Social Support and Quality of Life of the Child and Their Caregivers

For both groups, we found that greater perceived social support was associated with QoL. Nevertheless, for the group of children with DD, these associations were greater than in the group of children with TD.

The association between social support and quality of life is not a new idea in the literature. Fisher et al. (2022) [55] pointed out social support as a mediator of reduced stress and higher levels of life satisfaction. Accordingly, the availability of perceived social support seems to be directly linked to the greater opportunity for the caregiver to be in contact with friends [56] and have these people share their insecurity, uncertainties, and daily tasks. Previous studies reported similar results for other populations reporting that higher levels of social support were associated with better caregivers’ and children’s mental health [21].

Nevertheless, our results show us a very interesting point. It seems that the availability of perceived social support shows greater relevance for caregivers’ QoL than for children’s QoL. From these results, we can infer that greater levels of social support might help parents to deal with the challenge of raising their child, contributing to greater levels of caregiver QoL. Previous studies observed a positive relationship between perceived social support and parents’ QoL in families of children with DD [8]. Thus, it can be understood that parents who perceive the availability of social support usually show better QoL, which might contribute to improving their children’s well-being with emotional, social, and educational support even in a pandemic period. Caregivers of children with DD tend to be tired and can be frustrated and anxious [57]. Nevertheless, the support network can reduce the burden on these caregivers [58] because they can share their concerns and anxieties and receive help in daily activities, including the child’s home care.

It is also worth mentioning how perceived social support was much more related to the quality of life for caregivers of children with DD than for those with TD. In fact, the daily demands involved with raising a child with medical conditions are greater than the ones involved with raising typical children [59]. Therefore, for caregivers of children with DD, perceived social support seems to be more important than for caregivers with typical children since any available support has the potential to reduce the daily workload involved in raising these children. These results show us that even in a critical period, such as the pandemic, in which due to social distancing, interpersonal relationships may have been restricted in determining similar levels of social support for caregivers of children with DD.

## 5. Clinical Implications

The main clinical implications of our study come from our intriguing results. The peculiar features of social distancing and interruption of face-to-face activities seem to have changed the perceived social support for all families. Moreover, during the pandemic period, the unexpected finding of greater caregiver QoL levels for the DD group showed not only the resilience of these caregivers but also how much the perception and assessment of the quality of life in the face of a specific situation (e.g., COVID-19 pandemic. As mentioned by Pozniak and Kraus de Camargo (2021) [53], the silver lining of COVID-19 may be that it presents an opportunity to rethink existing social arrangements and rebuild a more equitable post-pandemic society.

Results point out also that even in this unprecedented scenario, the association between the perceived social support and quality of life of children with DD and their caregivers was clear. This association was especially evident for the caregiver’s QoL.

These results, obtained during a pandemic period, applied to a post-pandemic scenario, show how much the availability of social support for families of children with DD is relevant. Rehabilitation professionals providing family-centered therapy need to empower caregivers in the search for solid support networks which would contribute to higher levels of QoL. Future studies should investigate the main determinants of children’s QoL, ideally assessed directly instead of by proxy, in order to understand how to best improve it.

Considering we are now in a period where many of the previous restrictions have been lifted, it is important to keep attention on these aspects to enable the creation of strategies that favor improvements in QoL of the population, in general, but especially of children with DD and their caregivers.

## 6. Study Strengths and Limitations

Among the main limitations of our study, we may cite the assessment of children’s QoL by means of their parents’ reports. It is known that the reported QoL by parents might be lower than the reports provided by the children themselves [60]. Nevertheless, considering the pandemic scenario, which had allowed us to assess our participants remotely, a parents’ report was the best way we found to conduct the remote surveys. Another limitation was the absence of pre-pandemic data regarding children’s and caregivers’ social support and QoL, which would have provided us with the possibility to investigate COVID-19’s impact. Further data collection in a current post-pandemic scenario with a follow-up study would clarify to us the way families have been resuming their social support networks and the impact of this on their QoL and the QoL of their children.

A strength of the study is that it provides a unique view into the relationships between perceived social support and QoL during the “natural experiment” of living through a pandemic.

## 7. Conclusions

During the COVID-19 pandemic, the perceived level of social support for caregivers of children with DD and TD was similar, possibly reflecting social distancing effects. For children with DD, the caregiver-reported level of QoL in physical, psychosocial, and social activities was lower than in children with TD. Nevertheless, for their caregivers, the QoL levels were greater than the ones in the TD group. Finally, for both groups, greater levels of perceived social support were associated with greater caregiver-reported QoL of the child in the psychosocial domain and the caregiver, especially in the aspects of physical, emotional, social capacity, communication, concern, daily activities, and family relationships. These associations were more numerous, especially for the families of children with DD.

## Figures and Tables

**Figure 1 ijerph-20-04449-f001:**
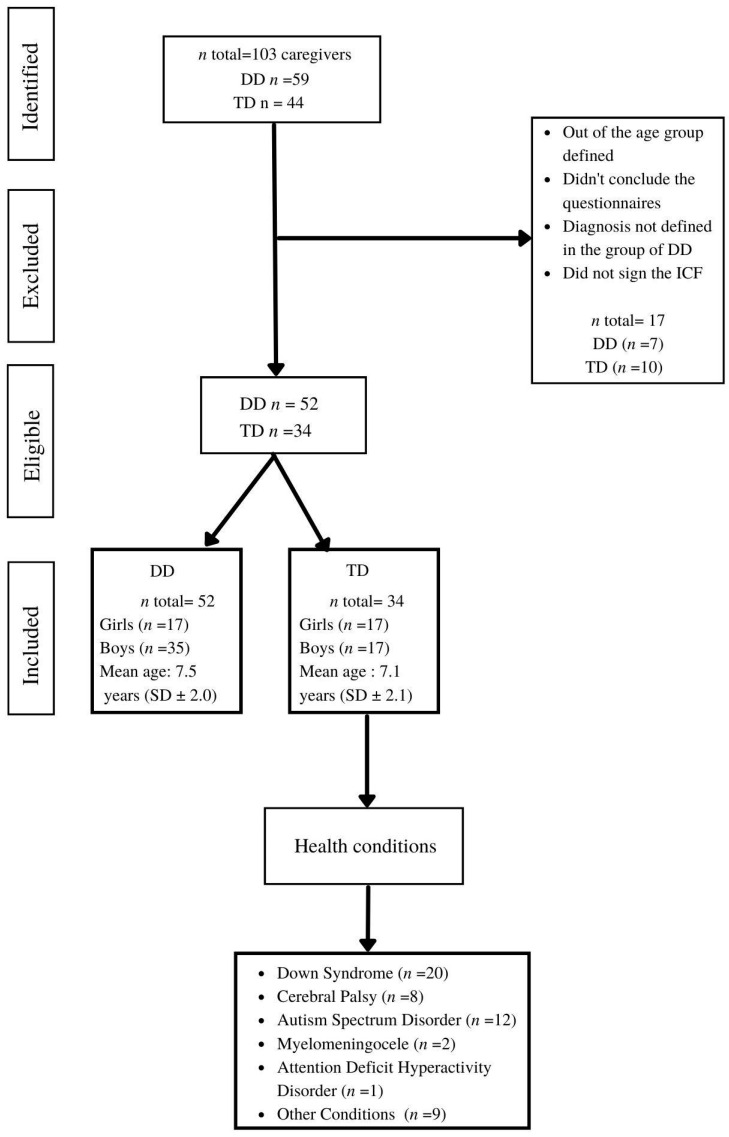
Flowchart of the participants. Legend: *n* = number of participants; DD = developmental disabilities; TD = typical development; SD = standard deviation; ICF = Informed Consent Form.

**Table 1 ijerph-20-04449-t001:** Characteristics of the sample of children with DD and TD and test of the difference between groups.

Children					
	Developmental Disabilities		Typical Development		Differences between Groups
	*n*	%	*n*	%	*p*-Value
Child’s Sex					
Female	17	32.70	17	50	0.158
Male	35	67.30	17	50	
Child’s Age (years)					
5	10	19.2	11	32.4	0.261
6	9	17.3	5	14.7	
7	9	17.3	7	20.6	
8	8	15.4	2	5.9	
9	6	11.5	3	8.8	
10	5	9.6	2	5.9	
11	3	5.8	3	8.8	
12	2	3.8	1	2.9	
Caregivers’ Sex					0.774
Female	52	100	33	97	
Male	0	-	1	3	
Caregivers’ Age (years)					0.087
<20	1	1.9	0	0	
20–30	8	15.4	6	17.6	
31–40	20	38.5	20	58.8	
41–50	17	32.7	8	23.5	
51–60	6	11.5	0	0	
Work situation					<0.001 *
Work (in person)	10	19.2	13	38.2	
Work (home office)	11	21.1	18	52.9	
Do not work	31	59.6	3	8.8	
Socioeconomic level					<0.001 *
A	4	7.7	8	23.5	
B1	4	7.7	7	20.6	
B2	15	28.8	13	38.2	
CI	17	32.7	5	14.7	
C2	10	19.2	1	2.9	
D–E	2	3.8	0	0	
Maternal schooling					<0.001 *
Incomplete Elementary	7	13.4	0	0	
Complete Elementary	3	5.7	0	0	
Incomplete high school	5	9.6	1	2.9	
Complete high school	15	28.8	2	5.8	
Incomplete Higher Education	2	3.8	1	2.9	
Complete Higher Education	20	38.4	30	88.23	

Legend: *n* = number of participants; * = *p* < 0.05.

**Table 2 ijerph-20-04449-t002:** Comparison between the children with DD and TD for the children’s QoL variables.

Children							
	Developmental Disabilities			Typical Development			
	Median (SD)	Minimum Value	Maximum Value	Median (SD)	Minimum Value	Maximum Value	*p*
Total PedsQL Child total	53.80 (16.18)	17.39	81.52	67.93 (13.10)	21.74	85.87	<0.001 *
PedsQL Child Psychosocial Health	55.0 (17.27)	15.00	86.66	75.0 (14.69)	20.00	91.67	<0.001 *
PedsQL Child Physical Health	53.12 (21.30)	9.37	90.62	67.19 (17.98)	31.25	100.00	<0.001 *
PedsQL Child Emotional Aspect	55.00 (20.49)	10.00	100.00	60.00 (18.75)	15.00	100.00	0.607
PedsQL Child Social Activities	52.50 (23.44)	0.00	90.00	85.00 (19.20)	10.00	100.00	<0.001 *
PedsQL Child School Activity	50.00 (19.21)	10.00	90.00	80.00 (16.92)	35.00	100.00	<0.001 *

Legend: SD = standard deviation; PedsQl = Pediatric Quality of Life Inventory version 4.0; * = *p* < 0.05.

**Table 3 ijerph-20-04449-t003:** Comparison between the children with DD and TD for perceived social support and caregivers’ QoL.

Caregivers							
	Developmental Disabilities			Typical Development			
	Median (SD)	Minimum Value	Maximum Value	Median (SD)	Minimum Value	Maximum Value	*p*
Perceived Social Support	66.0 (19.33)	31.00	95.00	74.5 (15.11)	47.00	95.00	0.096
Total PedsQL Family total	60.06 (17.08)	19.44	86.80	52.78 (16.07)	17.36	82.14	0.029 *
PedsQL Family Physical Capacity	70.83 (21.97)	8.33	91.67	56.25 (19.90)	12.50	91.67	0.006 *
PedsQL Family Emotional Aspect	65.00 (22.37)	10.00	100.00	45.00 (18.06)	10.00	90.00	0.001 *
PedsQL Family Social Aspect	62.50 (25.93)	6.25	100.00	40.25 (23.70)	6.25	93.75	0.001 *
PedsQL Family Mental capacity	70.00 (23.03)	10.00	100.00	65.00 (21.62)	0.00	100.00	0.170
PedsQL Family Communication	66.67 (28.36)	8.33	100.00	75.00 (21.99)	16.67	100.00	0.042 *
PedsQL Family Concern	40.00 (21.65)	0.00	100.00	35.00 (30.20)	0.00	100.00	0.448
PedsQL Family Daily Activities	50.00 (21.39)	8.33	100.00	33.33 (33.36)	0.00	100.00	0.004 *
PedsQL Family Family Relationships	70.00 (24.81)	0.00	100.00	60.00 (20.87)	0.00	100.00	0.086

Legends: SD = standard deviation; PedsQl = Pediatric Quality of Life Inventory Family Impact Module; * = *p* < 0.05.

**Table 4 ijerph-20-04449-t004:** Correlation between perceived social support with caregiver’s QoL and child’s QoL for the children with DD and TD and their caregivers.

Developmental Disabilities		Typical Development	
Variables	Perceived Social Support	Variables	Perceived Social Support
PedsQL Child *Total*	0.255	PedsQL Child *Total*	−0.78
PedsQL Child *Psychosocial Health*	0.350 * o	PedsQL Child *Psychosocial Health*	−0.026
PedsQL Child *Physical Health*	0.061	PedsQL Child *Physical Health*	−0.011
PedsQL Child *Emotional Aspect*	0.380 * o	PedsQL Child *Emotional Aspect*	−0.038
PedsQL Child *Social Activities*	0.233	PedsQL Child *Social Activities*	0.066
PedsQL Child *School Activity*	0.183	PedsQL Child *School Activity*	−0.34
Total PedsQL *Family total*	0.562 * #	Total PedsQL *Family total*	0.335
PedsQL Family *Physical Capacity*	0.402 * #	PedsQL Family *Physical Capacity*	0.121
PedsQL Family *Emotional Aspect*	0.492 * #	PedsQL Family *Emotional Aspect*	0.173
PedsQL Family *Social Aspect*	0.606 * #	PedsQL Family *Social Aspect*	0.472 * #
PedsQL Family *Mental Capacity*	0.249	PedsQL Family *Mental Capacity*	0.270
PedsQL Family *Communication*	0.535 * #	PedsQL Family *Communication*	0.431 * #
PedsQL Family *Concern*	0.303 * o	PedsQL Family *Concern*	−0.001
PedsQL Family *Daily Activities*	0.394 * o	PedsQL Family *Daily Activities*	0.325
PedsQL Family *Family Relationships*	0.369 * o	PedsQL Family *Family Relationships*	0.273

Legend: PedsQl = Pediatric Quality of Life Inventory; * = *p* < 0.05; o = weak correlation; # = moderate correlation.

## Data Availability

The dataset supporting the conclusion of this article is available from the authors upon reasonable request.

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
