# Peer review of "Perceived Social Support and Quality of Life of Children with and without Developmental Disabilities and Their Caregivers during the COVID-19 Pandemic in Brazil: A Cross-Sectional Study"

_ijerph, 2023, doi:10.3390/ijerph20054449_

Round 1

Reviewer 1 Report

Congratulations on reaching this stage of peer review on your project. I think there are some changes to make as you have not yet illuminated the originality of your findings - I hope these comments are useful in enabling you do make this more apparent. 

The finding that children do better in terms of psychosocial health in the context of social support is nothing new. You really need to explore the body of literature in relation to this, ie social inclusion and social groupings, and the impact of this on mental wellbeing, social skills and behaviours and social & emotional learning. Currently your study sits in an isolated position from this existing knowledge. This needs to consider both parents and children. 

Defining social support (4.2) - it would be better to explain this earlier on. 

Comments about the impact of covid would be better placed as Limitations (4.2).

"Families of children tend to be tired" rather reduces the point. You need to consult the literature about lived experience for caring for children with developmental delay. if you explain this more carefully, your points about social support will gain strength and be less reliant on the assumptions of readers. 

Some of the comments about family experience are under developed in context of other studies which have been done. For example, on p10 you state: Given this fact, they have more opportunities to maintain a regular social life and a healthier family relationship even during social distancing. It would be beneficial to draw on other studies so that you can understand your own findings in greater depth, or, offer your findings are original.   I am not convinced that it would have been possible for these families to maintain social support during covid restrictions - but it is important to consult the literature and find that out. 

Clinical implications - the last sentence is a little unclear - do you mean research should be carried out before during and after a future pandemic? If so, I am not sure how this would be viable.. 

Overall I think the quality of your write up needs to be enhanced by consulting the wider literature on: 1) the psychosocial impact of covid, 2) the psychosocial impact on social inclusion and connectedness for both parents and children, and related outcomes  3) the psychosocial issues for families where a child has DD. 

Good luck with your ammendments. 

Author Response

Reviewer #1:

Thank you so much for your valuable suggestions and for the opportunity to review our manuscript. All responses are highlighted in red in this letter as well as in the manuscript text. We made all changes you suggest, reviewing all the sections, specially introduction, methods and discussion, in order to clarify the current status of the literature, and the new insights our study is providing to the field. Also, we made a huge revision on the text/grammar in order to improve the manuscript. We hope the changes will meet your expectations.

The finding that children do better in terms of psychosocial health in the context of social support is nothing new. You really need to explore the body of literature in relation to this, ie social inclusion and social groupings, and the impact of this on mental wellbeing, social skills and behaviours and social & emotional learning. Currently your study sits in an isolated position from this existing knowledge. This needs to consider both parents and children.

Thank you for this important suggestion. We agree with you, and reformulated all the introduction section, adding new and more literature about the research topic.

Defining social support (4.2) - it would be better to explain this earlier on.

Thank you. We defined it in the introduction.

Comments about the impact of covid would be better placed as Limitations (4.2).

Thank you. You added this point in the limitation section.

"Families of children tend to be tired" rather reduces the point. You need to consult the literature about lived experience for caring for children with developmental delay. if you explain this more carefully, your points about social support will gain strength and be less reliant on the assumptions of readers.

We completely agree. Thank you. We added more literature about it in the introduction.

Some of the comments about family experience are under developed in context of other studies which have been done. For example, on p10 you state: Given this fact, they have more opportunities to maintain a regular social life and a healthier family relationship even during social distancing. It would be beneficial to draw on other studies so that you can understand your own findings in greater depth, or, offer your findings are original.   I am not convinced that it would have been possible for these families to maintain social support during covid restrictions - but it is important to consult the literature and find that out.

Thank you. We agree with you, and made a search on the literature about it. Thus, we added important knowledge about it in the introduction section. 

Clinical implications - the last sentence is a little unclear - do you mean research should be carried out before during and after a future pandemic? If so, I am not sure how this would be viable..

You are right! This sentence was not clear, and this is difficult to be viable. Because of that, we removed this part of the clinical implication, and added new, relevant and feasible aspects to clinical implication based on the insights our study provides.

Overall I think the quality of your write up needs to be enhanced by consulting the wider literature on: 1) the psychosocial impact of covid, 2) the psychosocial impact on social inclusion and connectedness for both parents and children, and related outcomes  3) the psychosocial issues for families where a child has DD.

THank you so much. We reviewed the literature about this topic, and reformulated introduction, methods, discussion, clinical implications, Study strengths and limitations and conclusions. We hope these changes will meet your expectations.

Reviewer 2 Report

In the summary, the wording should be modified, as it leads to confusion, especially in the section on material and methods, adapting what is stated to what is developed in the rest of the article. Modify the conclusion and write it in relation to the proposed objectives.

In the first paragraph of the introduction it talks about informal social support and formal social support. In the next sentence he quotes a study that analyses health-related social support. It should be explained beforehand what is meant by health-related social support and whether it can be considered formal or informal.

In the next paragraph you present studies that present the disorders that caregivers of children with developmental disabilities may experience.

In order to follow a correct line of argument, the introduction should start with the second and third paragraph and then explain what is meant by social support and the specific situation in Brazil.

In the paragraph preceding the statement of objectives, the following is quoted “ Furthermore, taking into account the prolongation of social restriction measures as-sociated with the fight against COVID-19, and its potential negative impact, it is believed to be relevant to study this association between families with children with TD and those with DD” What would be the benefit of knowing this relationship at the present time, since the end of the social pandemic restrictions months ago, needs to be discussed more concretely.

The same applies when citing as justification “Moreover, it is important to identify differences between groups with and without DD to guide intervention strategies oriented to these aspects, which are an important means by which the quality of life of the child and his/her caregivers can be favored” should be explained as a difference in the perception of social support and a subjective consideration of health status between these two groups would enhance the intervention.

In the next paragraph it states the expected results based on previous studies, which makes it clear that the expected results will not provide new knowledge or additional data. 

In the last sentence is stated “These results will contribute to a better understanding of these aspects and how changes in contextual factors can interfere with the QoL of caregivers and chil-dren with and without D”. This statement is not true, as in the methods section no instrument is used to measure changes in contextual factors, there is no pre/post study.

The justification for this research should be restructured. The objectives should be clearly defined and directly related to the novel contributions envisaged by the objectives.

In the section on clinical implications under discussion, an outdated situation on the current social constraints is presented.

The findings are in line with previous studies. The only original conclusion points to improved perception of social support by caregivers of children with developmental disabilities, but the discussion does not state how this can be reflected in improved social care or treatment.   

The whole article should be modified by establishing objectives, adequately justified by their clinical or social relevance, that seek novel results or that respond to a social or health demand, otherwise the article becomes a reproduction of previous articles whose development lacks solidity.

Author Response

Reviewer #2:

Thank you so much for your valuable suggestions and for the opportunity to review our manuscript. All responses are highlighted in red in this letter as well as in the manuscript text. We made all changes you suggest, reviewing all the sections, specially introduction, methods and discussion, in order to clarify the current status of the literature, and the new insights our study is providing to the field.  Also, we made a huge revision on the text/grammar in order to improve the manuscript. We hope the changes will meet your expectations.

In the summary, the wording should be modified, as it leads to confusion, especially in the section on material and methods, adapting what is stated to what is developed in the rest of the article. Modify the conclusion and write it in relation to the proposed objectives.

Thank you so much. We agree with you that the abstract was not telling about all we made on this research. Thus, we added a better explanation and details about it.

In the first paragraph of the introduction it talks about informal social support and formal social support. In the next sentence he quotes a study that analyses health-related social support. It should be explained beforehand what is meant by health-related social support and whether it can be considered formal or informal.

Thank you. We added this explanation you mention, in order to clarify the topic to the readers.

In the next paragraph you present studies that present the disorders that caregivers of children with developmental disabilities may experience.

In order to follow a correct line of argument, the introduction should start with the second and third paragraph and then explain what is meant by social support and the specific situation in Brazil.

Thank you for this suggestion. We agree with you about this sequence, and follow this in the new structure of our introduction.

In the paragraph preceding the statement of objectives, the following is quoted “ Furthermore, taking into account the prolongation of social restriction measures as-sociated with the fight against COVID-19, and its potential negative impact, it is believed to be relevant to study this association between families with children with TD and those with DD” What would be the benefit of knowing this relationship at the present time, since the end of the social pandemic restrictions months ago, needs to be discussed more concretely.

Thank you. We completely agree with you. We added this important aspect in the introduction.

The same applies when citing as justification “Moreover, it is important to identify differences between groups with and without DD to guide intervention strategies oriented to these aspects, which are an important means by which the quality of life of the child and his/her caregivers can be favored” should be explained as a difference in the perception of social support and a subjective consideration of health status between these two groups would enhance the intervention.

Thank you. We completely agree with you. We added this important aspect in the introduction.

In the next paragraph it states the expected results based on previous studies, which makes it clear that the expected results will not provide new knowledge or additional data.

Yes, you are right! Probably we did not explain our methods and the differential of our study through the manuscript sections. In order to improve this important aspect, we reformulated all introduction and discussion, and made improvements in all other sections.

In the last sentence is stated “These results will contribute to a better understanding of these aspects and how changes in contextual factors can interfere with the QoL of caregivers and chil-dren with and without D”. This statement is not true, as in the methods section no instrument is used to measure changes in contextual factors, there is no pre/post study.

Thank you for this comment. You are right, we did not compare pre/post pandemic, we only made analysis during the pandemic. We think this was not written in an understandable way, so we reformulated this part. The contextual factor  we were talking about was actually the pandemic period, and not contextual factors (related to the ICF), but since the sentence was not clear, It made it difficult to get the real meaning. So, we reformulated all this part and hope it will meet your expectations.

The justification for this research should be restructured. The objectives should be clearly defined and directly related to the novel contributions envisaged by the objectives.

Thank you. We made all these needs you suggested.

In the section on clinical implications under discussion, an outdated situation on the current social constraints is presented.

Thank you for this feedback. We agreed and reformulated all this part, in order to get a new and updated situation.

The findings are in line with previous studies. The only original conclusion points to improved perception of social support by caregivers of children with developmental disabilities, but the discussion does not state how this can be reflected in improved social care or treatment.  

Thank you. We believe that our manuscript gets insights about the pandemic period more than was written in the last version. Because of that, we reformatted all sessions in order to improve this.

The whole article should be modified by establishing objectives, adequately justified by their clinical or social relevance, that seek novel results or that respond to a social or health demand, otherwise the article becomes a reproduction of previous articles whose development lacks solidity.

We thank all the suggestions we made, and we believe now the manuscript is improved and can show the relevance of the findings and contribution to the literature. We hope these changes will meet your expectations.

Round 2

Reviewer 2 Report

The text has improved in quality and clarity but two aspects need to be changed before it can be published: 

1. the conclusion of the summary should be more specific. it is necessary to state the main conclusion of the article, which is currently very general. 

2. the specific data of the final sample and the table 1 should be removed. These data should be included in the results, not in the methodology. The methodology should describe the sample used for the study, the final data of the sample obtained is the result. 

Author Response

Reviewer #2:

Thank you for you second review. We appreciate the topics you highlight and we made all changes you suggest.  We hope the changes will meet your expectations.

  1. the conclusion of the summary should be more specific. it is necessary to state the main conclusion of the article, which is currently very general. 

We thank you your suggestions. In order to make more specific and complete the conclusion, we changed the conclusion of the summary:

Conclusion and Implications: During the COVID-19 pandemic, despite both groups presenting similar PSS, there are important differences in QoL between them. For both groups, greater levels of perceived social support are associated with greater caregiver-reported in some domains of the child’s and caregiver’s QoL. These associations are more numerous specially for the families of children with DD. This study provides a unique view into the relationships between perceived social support and QoL during the “natural experiment” of living through a pandemic.

  1. the specific data of the final sample and the table 1 should be removed. These data should be included in the results, not in the methodology. The methodology should describe the sample used for the study, the final data of the sample obtained is the result. 

Thank you for this suggestion. We moved this paragraph and the table 1 to the results section.
